

# A public database for the new MLST scheme for *Treponema pallidum* subsp. *pallidum*: surveillance and epidemiology of the causative agent of syphilis

Linda Grillova[1], Keith Jolley[2], David Šmajs[3] and Mathieu Picardeau[1]

[1] Biology of Spirochetes Unit, Institut Pasteur, Paris, France
[2] Department of Zoology, University of Oxford, Oxford, UK
[3] Department of Biology, Masaryk University, Brno, Czech Republic

## ABSTRACT

*Treponema pallidum* subsp. *pallidum* is the causative agent of syphilis, a sexually transmitted disease with worldwide prevalence. Several different molecular typing schemes are currently available for this pathogen. To enable population biology studies of the syphilis agent and for epidemiological surveillance at the global scale, a harmonized typing tool needs to be introduced. Recently, we published a new multi-locus sequence typing (MLST) with the potential to significantly enhance the epidemiological data in several aspects (e.g., distinguishing genetically different clades of syphilis, subtyping inside these clades, and finally, distinguishing different subspecies of non-cultivable pathogenic treponemes). In this short report, we introduce the PubMLST database for treponemal DNA data storage and for assignments of allelic profiles and sequencing types. Moreover, we have summarized epidemiological data of all treponemal strains ($n = 358$) with available DNA sequences in typing loci and found several association between genetic groups and characteristics of patients. This study proposes the establishment of a single MLST of *T. p. pallidum* and encourages researchers and public health communities to use this PubMLST database as a universal tool for molecular typing studies of the syphilis pathogen.

## INTRODUCTION

Syphilis is a bacterial infection caused by *Treponema pallidum* subsp. *pallidum*. Despite the availability of effective treatments, the annual worldwide incidence recently reached more than 5.6 million cases (*World Health Organization, 2016*).

The proper discrimination of strains of infectious pathogens is crucial for epidemiological and surveillance analyses, for description of population structure and dynamics and for the improvement of public health control strategies. More specifically, the association of specific strains of *T. p. pallidum* with different groups of patients can result in a better understanding of syphilis epidemiology. Association of macrolide resistance/sensitivity and allelic profiles are particularly important for a better

Corresponding author
Linda Grillova,
linda.grillova@pasteur.fr

understanding of the emergence of antibiotic-resistant strains. Finally, the molecular typing of syphilis-causing strains can help to determine syphilis diagnosis, especially in cases with atypical symptoms, thereby preventing syphilis-bejel confusion (*Grange et al., 2016*; *Noda et al., 2018*).

Multi-locus sequence typing (MLST) is typically based on amplification and sequencing of about seven housekeeping gene fragments (loci). For each locus, the different sequence variants are assigned as distinct alleles and, for each isolate, the combination of alleles defines the allelic profile and sequence type (ST). Allelic profiles and/or STs are used for definition of the strains and these can be clustered into clonal complexes based on shared alleles (*Maiden et al., 2013*). The traditional MLST approach is based on allele-based comparisons, where each allelic change is counted as a single event, regardless of the number of nucleotide changes involved. This helps to mitigate the effects of horizontal genetic transfer in organisms prone to recombination. The MLST data are commonly stored in publicly available and curated databases, such as PubMLST, which uses the Bacterial Isolate Genome Sequence Database (BIGSdb) platform (*Jolley & Maiden, 2010*). These databases can be used to automatically identify and assign new allele variants and STs, and provide tools to analyze the origin and phenotypic metadata of samples with the genotypic information of the typed pathogens. Several functions are available within this software to analyze and visualize epidemiological data (*Jolley & Maiden, 2010*; *Maiden et al., 2013*). BIGSdb is currently hosting over 100 different bacterial species and is one of the most widely used tools for epidemiological purposes (*Jolley, Bray & Maiden, 2018*).

Recently, we published a new MLST scheme of *T. p. pallidum* (*Grillová et al., 2018a*; *Pospíšilová et al., 2018*). The *T. p. pallidum* MLST scheme is somewhat unusual compared to typical MLST schemes reflecting the special characteristics of *T. p. pallidum*. The *T. p. pallidum* MLST uses three highly variable loci and performs sequence-based analyses, that is, it takes into account sequence differences among alleles.

i) Even though some recent attempts to cultivate *T. p. pallidum* were successful using *T. p. pallidum* strains passed through cultures with rabbit epithelial cells (*Edmondson, Hu & Norris, 2018*), the in vitro cultivation of *T. p. pallidum* strains isolated directly from patients is still not possible. It is, therefore, necessary to perform MLST directly from clinical samples, which usually contain a very low number of *T. p. pallidum* DNA copies (10–$10^2$ *T. p. pallidum* DNA copies/μl) (*Pinto et al., 2017*). To ensure a high proportion of fully typed samples, the newly proposed MLST is based on three loci only, in comparison to the conventional seven-locus scheme used in other bacteria.

ii) Multi-locus sequence typing schemes are usually based on several housekeeping genes. However, this approach is not suitable for monomorphic bacteria as *T. p. pallidum* (*Šmajs, Norris & Weinstock, 2012*; *Radolf et al., 2016*; *Šmajs, Strouhal & Knauf, 2018*). In this case, every single nucleotide variant (SNV) is important and valuable and the typing loci with the highest SNV density should be used to ensure sufficient variability to map the overall population diversity. However, the typing loci should still be stable over time. Therefore, the new MLST was designed on the principle of the

sequence analyses using three highly variable loci—TP0136, TP0548 and TP0705 (*Grillová et al., 2018a*). These three loci represented the lowest number of loci with the highest discrimination power possible among the candidate loci selected for typing.

iii) Because of the monomorphic character of *T. p. pallidum* genome and the low number of typing loci, the "sequence-based analyses" approach is highly recommended. This approach, compared to the conventional "allele-based comparisons," allows the researcher to distinguish alleles based on a number of nucleotide differences and permits interpretation of the data using maximum-likelihood phylogeny. Even though the *T. p. pallidum* MLST uses only three typing loci, the scheme has a high resolution (more than 30% of the resolution achieved with genome-wide data). This approach also allows distinction of *T. p. pallidum* from other treponemal subspecies and species, discriminates between the two *T. p. pallidum* clades (TP0136, TP0548) (*Nechvátal et al., 2014*), and can be used for differentiation of strains within each of these clades (TP0136 for the Nichols clade, and TP0705 for the SS14 clade).

The purpose of this short report is to establish a single universal MLST scheme applicable to *T. p. pallidum* (*Grillová et al., 2018a*) and introduce the PubMLST database for treponemal DNA data storage and ST assignments.

# MATERIALS AND METHODS

## *Treponema pallidum* subsp. *pallidum* database hosted at PubMLST

We have created a *T. p. pallidum* database hosted at PubMLST. PubMLST uses the BIGSdb platform for the storage and analysis of sequence data from bacterial isolates. The input format can be fasta files, which contain small contigs assembled from dideoxy chain termination sequencing of the typing loci, or complete or multiple contigs generated from next-generation sequencing platforms. When the new sequences are submitted to BIGSdb, the BLAST algorithm incorporated into BIGSdb identifies already known sequence variants or marks an unknown variant for curator verification. After verification by a curator, a novel allele number is assigned (*Maiden et al., 2013*). Metadata of the patients may be submitted to the database as well, and includes some important characteristics of isolates, for example, isolate identification name/number; country where the strain was isolated; biological source of sample; year of isolation; and resistance to macrolide antibiotics. However, we encourage the researchers to upload as much information about patients and samples as available (e.g., age, sex and sexual orientation of the patients, stage of the disease, serological results etc.). The database can be accessed at https://pubmlst.org/tpallidum/. The database is primarily designed to be used as a tool for automatic ST assignments; for determination of new alleles; for storage of sample metadata; for identification of new associations between genetic types and metadata using various tools; and for user-friendly visualization of molecular typing data, particularly using GrapeTree (*Zhou et al., 2018*) and Microreact (*Argimón et al., 2016*) plugins. These plugins enable, for example, visualization of genomic epidemiology and phylogeography by showing the correlation of the STs and geographical areas in real time and visualization of phylogenetic data together with patient metadata.

However, this database can also serve as a repository for complete/draft genomes and gene-based comparative genomic data (using the Genome comparator tool). Sequences and metadata can be exported from the database in multiple formats. The database is overseen by the curators, who will check the submitted data and who will be available for any additional help (see https://pubmlst.org/tpallidum/).

## Sequences submitted to PubMLST

All published whole genome and draft genome sequences of *T. p. pallidum* (*Pětrošová et al., 2012*, *2013*; *Giacani et al., 2010*, *2014*; *Zobaníková et al., 2012*; *Tong et al., 2017*; *Arora et al., 2016*; *Sun et al., 2016*; *Pinto et al., 2016*; *Strouhal et al., 2018*; *Grillová et al., 2018b*), data available at GenBank, as well as sequences obtained by MLST (*Grillová et al., 2018a*; *Pospíšilová et al., 2018*) were submitted to the database.

## Phylogenetic analyses

The phylogenetic tree was generated using MEGA6 (*Tamura et al., 2013*) with the maximum likelihood bootstrap algorithm and Tamura Nei model based on concatenated sequences of typing loci of fully typed samples ($n = 286$). The tree was visualized using iTOL (*Letunic & Bork, 2016*), available as an external plugin of the BIGSdb.

## Statistical analyses

Statistical analyses were performed using STATISTICA software v.12 (StatSoft, Tulsa, OK, USA) using data extracted from the two-field breakdown option. A Fisher's exact test was used to establish the correlations of patient's metadata and allelic variants and STs. Statistical significance was set at $p < 0.05$.

# RESULTS

## *Treponema pallidum* subsp. *pallidum* database

At the time of writing (September 2018), the databases contain 358 *T. p. pallidum* strains, mostly isolated from Europe (70.87%), followed by North America (26.33%), Asia (2.52%) and South America (0.28%) (Table 1). More than 90% of samples were clinically acquired and 31 strains were propagated in rabbits. A total of 71 samples represented whole/draft genomes and 287 samples were typed using MLST and yielded sequences only in typing loci. The samples were collected from 1912 to 2017. There were 272 samples collected from males (183 samples were collected from Men who have sex with men; MSM) and only four samples from females. Most of the samples were isolated from patients diagnosed with primary syphilis, followed by secondary syphilis, patients in the border of primary and secondary stage, and there were also three cases of congenital syphilis. The patients were between 0 and 71 years old. In addition, 21 samples were found negative by serology, but positive by PCR and all of them were isolated from patients with primary syphilis by genital, anal or throat swab (Table 1). Most of the samples (262, 73.39%) contained treponemes resistant to macrolide antibiotics caused by A2058G mutations (in 257 cases) and A2059G mutations (in five cases) in both treponemal 23S rRNA genes.

Fully typed *T. p. pallidum* strains ($n = 285$) were divided into 40 allelic profiles or strain types (ST1-ST40), and further into two clonal complexes, where 31 STs belong to the

**Table 1 Clinical characteristics of the *T. p. pallidum* strains submitted to the BIGSdb.**

| Continent (n, %) | | Clinical source (n, %) | | Stage (n, %) | |
|---|---|---|---|---|---|
| Europe | 253 (70.87) | Genital swab | 180 (50.42) | Primary | 119 (33.33) |
| North America | 94 (26.33) | Anal swab | 56 (15.69) | Primary/secondary | 8 (2.24) |
| Asia | 9 (2.52) | Throat swab | 39 (10.92) | Secondary | 40 (11.2) |
| South America | 1 (0.28) | Skin lesion | 17 (4.76) | Congenital | 3 (0.84) |
| **Country (n, %)** | | Blood | 3 (0.84) | Unspecified | 187 (52.38) |
| France | 146 (40.90) | CSF | 2 (0.56) | **Macrolide resistance (n, %)** | |
| Cuba | 72 (20.17) | Amniotic fluid | 1 (0.28) | Resistant | 262 (73.39) |
| Switzerland | 72 (20.17) | Tissue | 1 (0.28) | Sensitive | 62 (18.3) |
| Portugal | 25 (7) | Other | 1 (0.28) | Not done | 33 (9.24) |
| USA | 21 (5.88) | Unspecified | 57 (15.97) | **Mutations (n, %)** | |
| China | 9 (2.52) | **Sex (n, %)** | | A2058G | 257 (71.99) |
| Netherlands | 5 (1.4) | Male | 272 (76.19) | A2059G | 5 (1.4) |
| Austria | 3 (0.84) | Female | 8 (2.24) | Unspecified | 95 (26.61) |
| Czech Republic | 2 (0.56) | Unspecified | 77 (21.5) | **Serology (n, %)** | |
| Argentina | 1 (0.280 | **Sexual orientation (n, %)** | | Positive | 182 (50.98) |
| Mexico | 1 (0.28) | MSM | 183 (51.26) | Negative | 21 (5.88) |
| **Source (n, %)** | | MSW | 27 (7.56) | Unspecified | 154 (43.14) |
| Clinically acquired | 326 (91.32) | WSM | 3 (0.84) | | |
| Experimental animals | 31 (8.68) | Unspecified | 144 (40.34) | | |

Note:
CSF, cerebrospinal fluid; MSM, men who have sex with men; MSW, men who have sex with woman; WSM, woman who have sex with men.

clonal complex "SS14-like" and nine STs to the clonal complex "Nichols-like" (Fig. 1; Table 2). STs were divided based on the 137 variable sites in total present in the concatenated sequences of typing loci (2,584 bp), where 46 variable positions were found to be parsimony informative for distinction of the two clonal complexes. The majority of the samples were found to belong to the SS14-like clonal complex (92.3%), while only 7.7% of samples belonged to the Nichols-like clonal complex.

## Association found using *T. p. pallidum* BIGSdb

We have found several association between genetic data and metadata of the samples including patient characteristics. Nichols-like strains were found to be associated with susceptibility to macrolides, and, on the other hand, SS14-like strains were associated with the presence of mutations leading to macrolide resistance ($p < 0.0001$) (Fig. 2A). The data showed that Nichols-like strains appear to be spreading predominantly among MSM patients (Fig. 2B). Moreover, Nichols-like strains were associated with an older population (35 years old and older, $p = 0.0104$) (Fig. 2C). Most of these association were already described previously (*Grillová et al., 2014*; *Gallo Vaulet et al., 2017*; *Read et al., 2016*). However, in the future, there is a need to verify these association by examination of a higher number of treponemal strains from different geographical areas and from different groups of patients to avoid sampling biases.

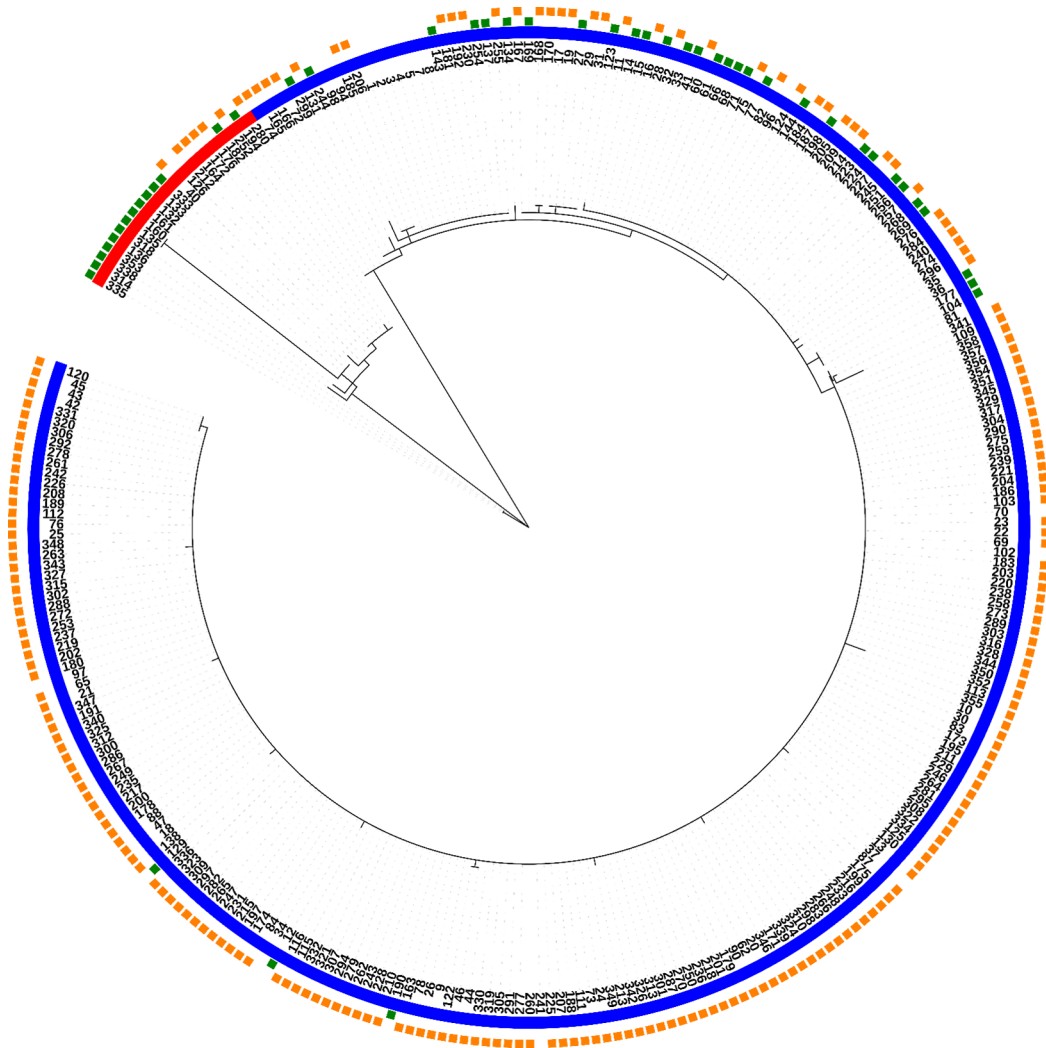

**Figure 1** **Phylogeny of all fully typed samples (*n* = 286) divided into 40 STs and two clonal complexes (Nichols-like and SS14-like), based on concatenated sequences of typing loci.** Blue color represents the SS14-like clade and red color represents the Nichols-like clade. The green squares mark the macrolide sensitive strains and the orange squares mark the resistant strains.

ST1 was found to be the predominant sequencing type (allelic profile: 1.3.1, clonal complex: SS14-like). However, with the increasing number of newly submitted *T. p. pallidum* strains, the number of different STs also increased, suggesting that the diversity of the *T. p. pallidum* strains is still not fully revealed. Interestingly, five STs were found exclusively among the strains propagated in rabbits—ST20–ST24 (*n* = 8) and not among the strains isolated directly from patients. However, the remaining examined strains propagated in rabbits (*n* = 23) shared the same STs with the directly sequenced strains.

## DISCUSSION

Since *T. p. pallidum* cannot be routinely cultivated in vitro, MLST needs to be performed directly from clinical samples. To obtain fully typed samples enabling assignment of ST,

**Table 2 Sequencing types found among 286 *T. p. pallidum* fully typed strains submitted to the BIGSdb.**

| ST | TP0136 | TP0548 | TP0705 | Clonal complex | No. of samples |
|----|--------|--------|--------|----------------|----------------|
| 1  | 1  | 3  | 1  | SS14-like     | 169 |
| 2  | 1  | 1  | 1  | SS14-like     | 37  |
| 3  | 1  | 1  | 8  | SS14-like     | 13  |
| 4  | 7  | 1  | 9  | SS14-like     | 5   |
| 5  | 2  | 1  | 2  | SS14-like     | 3   |
| 6  | 3  | 2  | 3  | Nichols-like  | 3   |
| 7  | 1  | 4  | 1  | SS14-like     | 3   |
| 8  | 1  | 5  | 1  | SS14-like     | 2   |
| 9  | 4  | 3  | 1  | SS14-like     | 1   |
| 10 | 1  | 11 | 8  | SS14-like     | 3   |
| 11 | 1  | 1  | 3  | SS14-like     | 4   |
| 12 | 5  | 3  | 8  | SS14-like     | 1   |
| 13 | 6  | 3  | 1  | SS14-like     | 1   |
| 14 | 1  | 8  | 1  | SS14-like     | 1   |
| 15 | 1  | 3  | 5  | SS14-like     | 1   |
| 16 | 1  | 3  | 7  | SS14-like     | 1   |
| 17 | 13 | 1  | 1  | SS14-like     | 1   |
| 18 | 1  | 9  | 1  | SS14-like     | 1   |
| 19 | 1  | 1  | 10 | SS14-like     | 2   |
| 20 | 10 | 14 | 10 | Nichols-like  | 3   |
| 21 | 1  | 13 | 10 | SS14-like     | 1   |
| 22 | 11 | 14 | 10 | Nichols-like  | 2   |
| 23 | 12 | 15 | 10 | Nichols-like  | 1   |
| 24 | 6  | 1  | 8  | SS14-like     | 1   |
| 25 | 1  | 26 | 1  | SS14-like     | 1   |
| 26 | 9  | 7  | 3  | Nichols-like  | 6   |
| 27 | 1  | 1  | 9  | SS14-like     | 2   |
| 28 | 1  | 17 | 9  | SS14-like     | 2   |
| 29 | 1  | 18 | 1  | SS14-like     | 1   |
| 30 | 1  | 19 | 1  | SS14-like     | 1   |
| 31 | 9  | 20 | 3  | Nichols-like  | 1   |
| 32 | 14 | 3  | 1  | SS14-like     | 1   |
| 33 | 1  | 1  | 11 | SS14-like     | 1   |
| 34 | 1  | 22 | 12 | SS14-like     | 1   |
| 35 | 1  | 23 | 1  | SS14-like     | 1   |
| 36 | 1  | 1  | 13 | SS14-like     | 1   |
| 37 | 15 | 7  | 3  | Nichols-like  | 4   |
| 38 | 9  | 24 | 8  | Nichols-like  | 1   |
| 39 | 9  | 25 | 3  | Nichols-like  | 1   |
| 40 | 16 | 3  | 1  | SS14-like     | 1   |

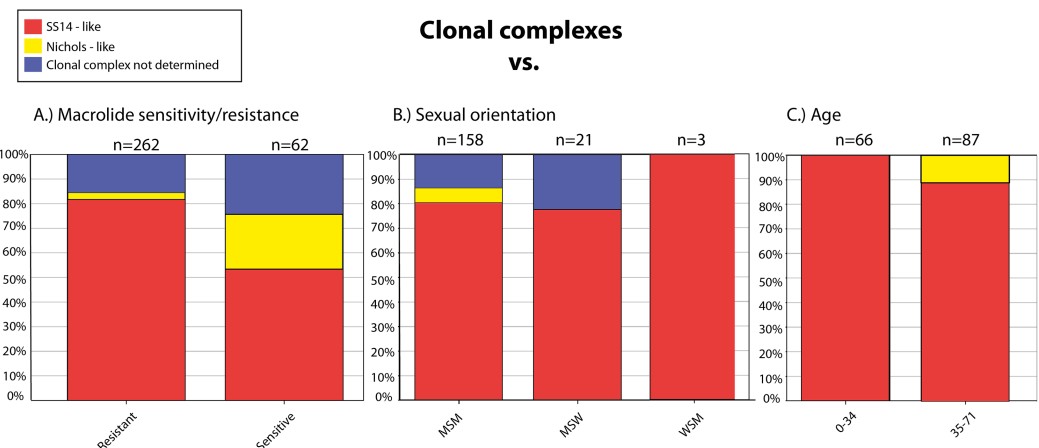

**Figure 2 Clonal complexes associated with the macrolide sensitivity/resistance (A), sexual orientation (B), and age of the patients (C).** The figure was created based on the BIGSdb output.

the amplification efficiency of all three loci needs to be as high as possible. The amplification efficiency depends on several factors, including the type of material taken from patients, the time between sampling and DNA isolation, the type of DNA isolation protocol as well as the type of amplification protocol.

As reported in many studies, the highest concentration of the treponemal DNA is present in swabs taken from the primary chancre, followed by blood-derived samples, cerebrospinal fluid and others (*Peng et al., 2011*; *Pinto et al., 2017*). For example, the swabs taken from primary chancre represent the samples with the highest concentration of treponemal DNA and are the most suitable candidates for MLST. However, MLST profiles revealed from parallel samples taken from the same patient (e.g., primary chancer swab and whole blood) are also important with respect to the stability of the used typing loci. For instance, the typing loci used in the enhanced CDC-typing (based on restriction fragment length polymorphism analyses of the *tprE*GJ genes and a number of repeats in the *arp* gene (*Marra et al., 2010*) were identified to be genetically unstable (*Mikalová et al., 2013*). This genetic instability was shown by revealing the different subtypes in samples isolated from swab and whole blood, both taken from the same patient. Loci selected for MLST (TP0136, TP0548 and TP0705) are considered to be genetically stable, showing the same allelic profiles in parallel samples as well as in samples collected from different rabbit passages (*Grillová et al., 2018a*).

Another important factor to obtain high-quality DNA is the time between sampling of patients and the DNA isolation. This time period should be as short as possible. If immediate DNA isolation is not possible, the swab extracts (PBS) should be stored at −20 °C. The swab extracts should be centrifuged at the lowest possible speed for 10 min and the supernatant should be used for the subsequent isolation of DNA. Even though it is time consuming, nested PCR should be used for target loci amplification to achieve the highest amplification efficiency. Moreover, we recommend looking for the presence or absence of A2058G and A2059G mutations in 23S rDNA, which encodes the macrolide resistance (*Grillová et al., 2018a*).

Most of *T. p. pallidum* strains analyzed to date from experimentally infected rabbits belong to the Nichols-like group, while most clinical *T. p. pallidum* isolates obtained directly from patients belong to the SS14-like group (*Šmajs et al., 2016*). We encourage researchers and public health communities to use MLST, which is able to distinguish these two clonal complexes. Moreover, we recommend submitting the sequencing data to PubMLST along with the metadata (including data from the serological tests). The expansion of the sample set in the PubMLST with syphilis isolates from different geographical areas can lead to a better understanding of the epidemiology of the two genetically distinct groups of syphilis treponemes.

## CONCLUSIONS

Here, we present the PubMLST BIGSdb database of *T. p. pallidum*, which represents the first publicly available data storage of *T. p. pallidum* sequences connected to metadata of patients. The database identifies specific regions of uploaded sequencing data and their genetic variation in order to reveal the sequencing types of *T. p. pallidum* isolates based on the newly developed MLST (*Grillová et al., 2018a*). We hope that this tool will open new opportunities in epidemiology of syphilis thereby allowing studies of sequencing types in different locations, tracking of syphilis infections and finding the association of particular strains with specific groups of patients. All these data should improve our understanding of syphilis epidemiology.

## ACKNOWLEDGEMENT

We thank to Dr. Robert Anthony Gaultney for English editing.

### Funding

Linda Grillova was funded by a PTR grant (PTR30-17) from the Institut Pasteur. This work was supported by the Grant Agency of the Ministry of Health of the Czech Republic (17-31333A) to David Šmajs. The development of PubMLST and BIGSdb has been supported by a Wellcome Trust Biomedical Resource Grant (104992). The funders had no role in study design, data collection and analysis, decision to publish or preparation of the manuscript.

### Grant Disclosures

The following grant information was disclosed by the authors:
Institut Pasteur: PTR grant: PTR30-17.
Grant Agency of the Ministry of Health of the Czech Republic: 17-31333A.
Wellcome Trust Biomedical Resource Grant: 104992.

### Competing Interests

The authors declare that they have no competing interests.
## Author Contributions

- Linda Grillova conceived and designed the experiments, performed the experiments, analyzed the data, prepared figures and/or tables, authored or reviewed drafts of the paper, approved the final draft.
- Keith Jolley conceived and designed the experiments, performed the experiments, analyzed the data, contributed reagents/materials/analysis tools, authored or reviewed drafts of the paper, approved the final draft.
- David Šmajs authored or reviewed drafts of the paper, approved the final draft.
- Mathieu Picardeau authored or reviewed drafts of the paper, approved the final draft.

## Data Availability

Raw data are available at PubMLST: https://pubmlst.org/tpallidum/

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
