# Peer review of "A public database for the new MLST scheme for Treponema pallidum subsp. pallidum: surveillance and epidemiology of the causative agent of syphilis"

_PeerJ, doi:10.7717/peerj.6182_

## Round 0.1 · original submission · Minor Revisions

As indicated below in the provided reviews, only minor modifications to the text are required before this manuscript is acceptable for publication.

Reviewer 1 ·

Basic reporting

Line 40-43. This statement is unclear. Please rephrase it.
Line 65-68. The in vitro cultivation of laboratory strains of T. pallidum using rabbit epithelial cells for up to 100 generations is a major breakthrough and it appears this method may support cultivation of strains directly from clinical specimens. Unless the authors can justify, “…we are still far away from the in vitro cultivation..”, this statement should be rephrased.
Line 73-75. Are there other examples of “monomorphic bacteria” where fewer MLST targets were better for genotyping versus conventional MLST as alluded to by the authors?
Line 175. The statement “Since TPA is not cultivable in vitro…” is not entirely correct and should be rephrased to reflect clinical strains cannot be routinely cultivated in vitro therefore, MLST or other typing methods have to be applied directly on clinical specimens.
Line 189-190: Does the “genetic variability” possibly reflect the low copy numbers of T. pallidum often seen in blood versus swabs? Could this account for preferential amplification of the tpr genes targets?
Line 194. This statement seems out of place and it does not flow from the previous paragraph.
Figure 2. Does no value mean the clonal complex was not determined?

There are several grammatical and syntax errors in the manuscript that need to be corrected.
Line 111: change “primary” to “primarily”

Experimental design

No comment

Validity of the findings

No comment

Additional comments

The authors describe a public database for the recently proposed MLST typing system for T. pallidum. As with other MLST schemes, this is an important tool for assigning strain types and will avoid confusion when new strain types are encountered by different groups.

Reviewer 2 ·

Basic reporting

No comment

Experimental design

No comment

Validity of the findings

No comment

Additional comments

The manuscript “A public database for the new MLST scheme for Treponema pallidum subsp. pallidum: surveillance and epidemiology of the causative agent of syphilis” by Linda Grillova and colleagues focused on an important topic about the molecular typing schemes for Treponema pallidum subsp. pallidum. In this study, it introduced a harmonized typing tool for consideration of several different molecular typing currently available and described the PubMLST database for treponemal DNA data storage and for assignments of allelic profiles and sequencing types. It would be benefit for molecular typing studies of the syphilis pathogen in the future. But there are some issues which should be addressed:
1) On line 123-124, Ref Grillová et124 al. 2018, it would be not appropriate. It was not the original research on the whole genome sequencing of Treponema pallidum.
2) On line 141, “passed” would be better change for “propagated”; Also on Line 172.
3) On line 155-157, “STs were divided based on the 137 variable sites present in the concatenated sequences of typing loci (2584 bp), where 46 variable positions were found to be parsimony informative for distinction of the two clonal complexes.” The variable site was relative to which reference? Nichols or SS14 ? It would be better described more clearly. And for the 137 variable sites, there would be good to present a summary table to show the detailed information about this position.
4) As the paper described, there was included 31strains propagated by rabbits, but on line 172-173, “Interestingly, five STs were found exclusively among the strains passed through rabbits-ST20-ST24.” It was only 8 strains in ST20-ST24. This section should contain a series of contradictory statements. And the authors mentioned that the phenomenon was interesting, what's suggested imply of this phenomenon?

·

Basic reporting

The authors are to be congratulated for their work to generate a central database for T.pallidum. As the sequencing of this organism grows such approaches may have increasing value.

I have only minor comments

I would have liked to see a more clearly articulated argument about what the database adds to the existing availability of the data/sequences through existing portals. I believe this argument can be made but would like to see if more clearly put forward.

Line 147 - I am not sure it is particularly surprising that a small number of samples were PCR positive/serology negative - this is a well recognised clinical phenomenon. Suggest re-word this line.

Experimental design

N/A

Validity of the findings

I found the section on associations to be slightly out of keeping with the rest of the manuscript. In particular there is limited/no discussion of the extent to which associations found may be reflective of sampling bias - as the authors themselves are aware there has been limited geographic, epidemiolgical range in sequencing of T.pallidum thus far and the extent to which associations reflect true relationships vs sampling artefacts should be discussed.

---

## Round 0.2 · accepted · Accept

Thank-you for your rapid response to the reviewer comments. I am happy to now be able to accept your manuscript for publication.

#